

# Assessing the Impacts of Hydrologic and Land Use Alterations on Water Temperature in the Farmington River Basin in Connecticut

John R. Yearsley[1], Ning Sun[2], Marisa Baptiste[3], Bart Nijssen[1]

[1]Department of Civil and Environmental Engineering, University of Washington, Seattle, 98195, USA
[2]Energy and Environment Directorate, Pacific Northwest National Laboratory, Richland, 99352, USA
[3]King County Department of Natural Resources and Parks, Seattle, 98104, USA

*Correspondence to*: John Yearsley (jyearsle@uw.edu)

**Abstract.** Aquatic ecosystems can be significantly altered by the construction of dams and modification of riparian buffers and the effects are often reflected in spatial and temporal changes to water temperature. To investigate the implications for water temperature of spatially and temporally varying riparian buffers and dam-induced hydrologic alterations, we have implemented a modeling system (DHSVM-RBM) that couples a spatially distributed land surface hydrologic model, DHSVM, with the distributed stream temperature model, RBM. The basic modeling system has been applied previously to several similar-sized watersheds. However, we have made enhancements to DHSVM-RBM that simulate spatial heterogeneity and temporal variation (i.e. seasonal changes in canopy cover) in riparian vegetation, and we included additional features in DHSVM-RBM that provide the capability for simulating the impacts of reservoirs that may develop thermal stratification. We have tested the modeling system in the Farmington River basin in the Connecticut River system that includes varying types of watershed development (e.g. deforestation and reservoirs) that can alter the streams' hydrologic regime and thermal energy budget. We evaluated streamflow and stream temperature simulations against all available observations distributed along the Farmington River basin. Results based on metrics recommended for model evaluation compare well to those obtained in similar studies. We demonstrate the way in which the model system can provide decision support for watershed planning by simulating a limited number of scenarios associated with hydrologic and land use alterations.

## 1 Introduction

Aquatic ecosystems can be significantly altered by the construction of dams ((Bednarek, 2001; Brooker, 1981; Lessard and Hayes, 2003; Ligon et al., 1995; Magilligan and Nislow, 2001; Olden and Naiman, 2010; Power et al., 1996; Preece and Jones, 2002; Yang, 2005) and modification of riparian buffers (Chen and Carsel, 1998; Gomi et al., 2006; McGurk, 1989; Moore et al., 2005; Sun et al., 2015). The effects of these alterations are often reflected in spatial and temporal changes to stream temperatures. It is well known that stream temperature has a major role in the function of aquatic ecosystems (Poole and Berman, 2001), and because of this, enhancing riparian vegetation has become an important part of stream restoration projects



and water quality management plans (e.g. Oregon Department of Environmental Quality, 2016). Similarly, construction and operation of impoundments and reservoirs for flood control, hydroelectric power generation and water supply can have significant impacts on stream temperatures (Bednarek, 2001).

The state-space paradigm (see Supplementary Material) provides a broad class of mathematical methods for state estimation that includes estimation for static linear systems, discrete-time linear dynamic systems, continuous-time linear dynamic systems, static nonlinear systems, and discrete-time nonlinear systems (Schweppe, 1973). State space estimators of stream temperature, for which the matrices in Equations S.1 and S.2 have been parameterized from first principles within the framework of the conservation laws governing momentum and energy (Boyd and Kasper, 2007; Chapra et al., 2008; Cole and Wells, 2002; Pike et al., 2013; Yearsley, 2009), provide a general approach for simulating non-stationary natural and anthropogenic processes. These model constructs are sufficiently general to accommodate a broad spectrum of time and space scales; ones that are based on the thermal energy budget for streams and rivers. They often come with a price, however, because estimating the parameters can become an exercise in solving a highly underdetermined geophysical data analysis problem. This is a problem that depends on time and space scales and model complexity. In the case of model development for complex aquatic ecosystems at high frequencies and small scales, some have approached these issues by limiting spatial and temporal scales and by designing rigorous and extensive data collection programs (Boyd and Kasper, 2007; Glose et al., 2017). State-space models of water temperature, developed at regional and global scales, have a higher level of aggregation for spatial scales and include only the most important components of the thermal energy budget (van Vliet et al., 2012; Wu et al., 2012; Yearsley, 2012, 2009).

Working within the broad framework of the conservation of momentum and thermal energy, here we investigate the way in which an integrated state-space model system of hydrology and water temperature (DHSVM-RBM) can be used to characterize the impacts of landscape alteration at intermediate spatial and temporal scales. We do so using a limited number of field observations and relying to a greater degree on data that are readily available from multiple sources rather than from monitoring programs designed specifically for our purposes. Furthermore, we can rely on the robust nature of the temperature response of streams and rivers to environmental forcings.

Using this approach, our goals are to: (1) develop and evaluate the nominal solutions for state-space estimates of streamflow and water temperature in the Farmington River basin of Connecticut using methods derived from physical principles and data gleaned from publicly-available repositories, and (2) use our model to provide examples from the Farmington River basin of the way in which the models can be used to evaluate water temperature impacts of changes in riparian vegetation and water storage.

## 2 Background and Site Description

The Farmington River basin (Figure 1) has an area of 1,559 km$^2$ with two main branches, the West Branch and the East Branch. The West Branch is 129.4 km in length and has its source in Hayden Pond near Otis, Massachusetts, while the East Branch is



75.2 km in length with headwaters in the Barkhamstead Reservoir near New Hartford, Connecticut. Land cover for the Farmington River basin, according to National Land Cover Database 2011 (NLCD 2011) is characterized by 58% of forest and 24% of developed land with the remaining 18% comprised of wetlands (8%), farmland (7%), and open water (3%). Seventy-three percent of the soils in the Farmington River basin are sandy loams according to the US Department of

Agriculture (USDA) Natural Resources Conservation Service (NRCS) States Soil Geographic (STATSGO).

As was true for the entire Connecticut River basin (Cogbill et al., 2002), deforestation began in the Farmington River basin in the 18th century as European settlers cleared land for agriculture and timber harvesting (Wharton et al., 2004). However, as a result of the decline of the agricultural industry in the State of Connecticut, reforestation began in the later part of the 19th century. More recently, forestlands have decreased again as a result of forest

harvest and urban growth. Jeon et al. (2014) reported that the annual rate of loss of forestlands due to permanent changes in land use in Connecticut was 1321 ha/year from 1990-2000 and 979 ha/year from 2000-2005, particularly in the heavily urbanized counties of Connecticut (Fairfield and New Haven counties). Nevertheless, about 60% of Connecticut remains in forestland, including the counties of Litchfield and Hartford comprising the Farmington River basin (75% and 53% forested, respectively; Wharton et al., 2004).

In addition to landscape alterations due to forest transition, the Farmington River basin has experienced considerable development of water management projects for purposes of water supply, flood control, hydroelectric power generation and recreation. The study of the Farmington River watershed by the Farmington River Watershed Association (2004) reported a total of 409 impounded lakes and reservoirs in the basin. They described most of these as smaller-scale impoundments, many of which were constructed in the early 1900's and are in disrepair or

no longer necessary. There are, however, a number of impoundments developed for water supply, recreation, flood control or generation of hydroelectric power that are of sufficient size to have an impact on the aquatic ecosystem of the river basin. For the purposes of this study, we chose to limit our study to seven large reservoirs (Table 1) with the potential for significantly modifying the water temperature regime of the natural system.

**3 DHSVM-RBM Model Identification**

DHSVM-RBM (Sun et al., 2015) integrates the physics-based Distributed Hydrology Soil Vegetation Model (DHSVM; Wigmosta et al., 1994), modified to include a riparian shading module (Sun et al., 2015), into the vector-based version of the semi-Lagrangian stream temperature model RBM (Yearsley, 2009) extended with a module that simulates certain basic features of deep reservoirs.



### 3.1 Hydrology and Riparian Shading Model

DHSVM was developed originally to simulate the hydrologic cycle in mountainous terrain (Wigmosta et al., 1994) based on saturation excess mechanisms. Subsequent adaptations have extended the capability of DHSVM to include urban landscapes with impervious surfaces and runoff detention (Cuo et al., 2008), glacier hydrology (Naz et al., 2014), riparian shading (Sun

et al., 2015), urban water quality (Sun et al., 2016), forest-snow interactions in canopy gaps (Sun et al., 2018), and reservoir regulation (Zhao et al., 2016). For this study, the riparian vegetation module of DHSVM described in Sun et al. (2015) has been modified to account for spatial and temporal variation in riparian vegetation. The spatial variations of riparian vegetation along the longitudinal axes of river segments are characterized by varying riparian vegetation parameters on a stream segment-by-segment basis. The temporal variability of the effects of riparian vegetation is described by a monthly-varying extinction

coefficient in the Beer's law representation of radiation transmission through the canopy (Wilson and Baldocchi, 2000).

DHSVM simulates the water and energy budget in the Farmington River basin at three-hourly time scales and 150-meter spatial scales for the physical processes of canopy interception, evaporation, transpiration and runoff generation driven by spatial input of soil, vegetation, topography and climate. Gridded meteorological inputs consist of precipitation, air temperature, downward shortwave and longwave radiation, wind speed, and relative humidity (Table 2). The 3-hourly

meteorological forcing data were disaggregated, using MTCLIM (Bohn et al., 2013; Thornton and Running, 1999), from a daily forcing dataset for the Conterminous United States developed by Livneh et al. (2013). We used the DEM from the US Geological Survey (USGS) National Elevation Dataset (NED) rescaled from 1 arc-second (app. 30 m) to 150 m. The soil class map was taken from STATSGO. The land cover data was taken from the USGS NLCD for Year 2011 rescaled from 30 m to 150 m.

We derived the riparian vegetation parameters by aggregating estimates of tree height, buffer width, bank-to-canopy distance and stream width in 1000-meter segments using online tools in Google Maps. We estimated tree height by comparing the height of trees with cultural features such as buildings or utility poles using Google Maps Browse Street View. We incorporated time-dependent leaf-area indices from Wilson and Baldocchi (2000) into the riparian shading model to characterize the monthly variability of the deciduous forest stands. We treated the riparian vegetation parameters estimated in this way as the

baseline condition, representative of existing conditions in the watershed.

### 3.2 Stream and Reservoir Temperature Models

RBM is a one-dimensional, time-dependent model of water temperature for streams and rivers dominated by advection that are well mixed vertically and laterally. RBM employs a semi-Lagrangian particle-tracking numerical scheme that has proven to be accurate and efficient in riverine systems dominated by advection. The numerical scheme is highly scalable in time and

space and has been applied to study the water temperature responses to changing climate, land use and riparian vegetation at the watershed and regional scales (Cao et al., 2016; Sun et al., 2015).



For those stream and river segments dominated by advection, we follow the method for obtaining the nominal solution to the one-dimensional, time-dependent equations for the thermal energy budget (Yearsley, 2012, 2009) (see Supplementary Materials). Initial conditions are required for solving difference equations with the form of Equation S.5 (see Supplementary Materials). Errors in estimating initial conditions will introduce uncertainty into the solution only for the time it takes for the

particle of water to traverse from its headwaters to its outlet. For example, the West Branch is the longest (129 km) branch of the Farmington River. Assuming a moderate stream speed of 0.5 meters/second, the time of travel associated with an initial parcel in the West Branch would be about three days. Since our simulations span a period of several months, we have assumed the initial water temperature, throughout the Farmington River to be 0.0 °C, and that errors associated with initial conditions have been dissipated after a spin-up time that is greater than the travel time of a water parcel from the headwaters to the

confluence with main stem Connecticut River.

Headwaters temperatures are the temperatures of the water that enter the surface water at the stream source and result from rainfall, snow melt or hyporheic flow. They have an important role in the heat budget of stream and rivers, the extent of which depends on the nature of the watershed (Janisch et al., 2012). However, their influence on downstream water temperatures diminishes as a function of stream speed, stream depth and meteorological forcing, reducing the uncertainty associated with

errors in estimates as water parcels travel downstream (Yearsley, 2012). Here, as in previous studies (Sun et al., 2015; Wu et al., 2012; Yearsley, 2012, 2009), we used the nonlinear regression of the logistic relationship between air temperature and water temperature from Mohseni et al. (1998). We estimated the parameters, $\alpha$, $\beta$, $\gamma$, and $\mu$, in Equation S.8 (see Supplementary Material) by minimizing the sum of the squared difference between Equation S.8 and the water temperature at CTDEEP-15844 (Figure 1). We further assumed that the estimates of the parameters for this location could be extended to the

other headwaters in the Farmington River basin.

The reservoir module added to the RBM software represents a simplified approach to characterizing those river segments with reservoirs having long residence times and potential for vertical stratification of water temperature. Based on their volume and inflow rate, we included those reservoirs on significant tributaries that had the potential for developing thermal stratification. Reservoirs included in our study with these characteristics are shown in Table 1. We obtained the reservoir geometry and

operating characteristics for those segments in the Farmington River basin with longer residence times (Table 1) from the NID USACE (2015) and Julian et al. (2013). Our approach, similar to that described in Boehlert et al. (2015), Chapra (1997), and Niemeyer et al. (2018), conceptualizes river segments with reservoirs that may stratify as two layers, a surface layer, the epilimnion and a bottom layer, the hypolimnion. Those stream segments identified as having a reservoir with potential for stratification were treated as multiple cells, depending on reservoir length, for purposes of estimating the flux of thermal energy

across the air-water interface and advective tributary input. For reservoirs with more than one cell, the cells were combined after computing surface transfer of thermal energy and advective tributary input into a single well-mixed epilimnion and single well-mixed hypolimnion CSTR's as in Figure 2. Specific details of the development of the nominal solutions for both the




stream segments dominated by advection and those that may develop an epilimnion and hypolimnion are given in the Supplementary Material section.

For the purposes of this study only, we assumed that the volumes of the epilimnion and hypolimnion were equal and were constant for the duration of the simulation period and that the outflow from each reservoir was equal to the total inflow. For those reservoirs whose primary function is water supply (Table 1), all outflow was from the epilimnion. In the case of Lake Colebrook, which has a controlled hydraulic sluice and hydropower turbines near the reservoir bottom, we assumed that all outflow was from the hypolimnion. Tributary inflow was placed into either the epilimnion or hypolimnion depending on whether the density of the inflow was lesser or greater than the density of the epilimnion, as in Niemeyer et al. (2018).

There are also lakes and impounded segments in the Farmington River with shallow depths and short residence times. We simulated the water temperature in these segments assuming they could be treated as cross-sectionally-averaged, well-mixed segments. Furthermore, we assumed that all these reservoirs had constant surface elevation and that the speed of the water parcel $U(n\Delta,x_j)$ at time $n\Delta$ and location $x_j$, could be estimated from the continuity equation. Stream speeds for those reservoirs in the Farmington River basin described as "River Run" in Table 1 were estimated in this way.

Within the coupling scheme, DHSVM provided DHSVM-RBM hydrologic and meteorological forcing input for each river segment, which are aggregated as the length-weighted average of the gridded 150 m cells that intercept a stream segment as in Sun et al. (2015). RBM makes use of DHSVM stream network file to establish the topology for its numerical solution technique. Stream networks, created in DHSVM with DEMs and ESRI Arcinfo macro language (AML) scripts, are composed of shorter stream segments connected within a network, with unique attributes of length, width, orientation, and Strahler stream order. Stream segment connectivity was used to define the network topology for DHSVM-RBM's particle tracking scheme.

We used inflow and outflow to each segment in the stream network simulated by DHSVM to estimate (1) the depth, $D(n\Delta, x_j)$, in one-dimensional, time-dependent formulation of the thermal energy budget (Equation S.14 (see Supplementary Materials) , and (2) the stream speed, $U(n\Delta, x_j)$, in the particle-tracking algorithm (Equation S.6 (see Supplementary Materials) with the method from Leopold and Maddock (1953). We estimated the parameters, $d_a$, $d_b$, $u_a$ and $u_b$, in Equations S.7 and S.8 (see Supplementary Materials) for the freely flowing segments RBM from the rating curve/stage-discharge relationships at the USGS gage sites shown in Table 3. RBM accommodates spatial variability for the coefficients that describe stream speed and depth as a function of streamflow. However, there are only a limited number of gages from which to estimate their values. We then assumed the parameter estimates at the individual gages could be extended to other stream segments as shown in Table 3.

### 3.3 Model Evaluation

State estimation techniques are applied to a broad spectrum of environmental fields and there has been considerable effort to develop approaches for evaluating model performance. In general, these efforts recognize the many aspects to assessing model





performance and attempt to identify appropriate metrics for doing so (Bennett et al., 2013; Moriasi et al., 2007; Krause et al., 2005; Willmott, 1982). Moriasi et al. (2007) have recommended a qualitative rating system for evaluating model results at daily, monthly and annual time steps at watershed scale based on a meta-analysis of several applications of the watershed models, e.g. Soil & Water Assessment Tool (SWAT) (Neitsch et al., 2011), the Hydrological Simulation Program-FORTRAN

(HSPF) (Bicknell et al., 1997) and WARMF (Herr and Chen, 2012). Performance measures for streamflows in these recommendations include the Nash-Sutcliffe Efficiency (NSE), the coefficient of determination ($r^2$) in conjunction with gradient and intercept of the regression line, the root-mean-square error (RMSE) and the percent bias (PBIAS). The hydrologic model, DHSVM, simulates streamflow at the length and time scales similar to SWAT, HSPF and WARMF and we used these same performance measures at daily and monthly time steps to evaluate the results in our study.

There has been no similar meta-analysis of performance measures for stream temperature. However, most studies of stream temperature provide estimates of one or more of the same performance measures (Cuo et al., 2008, 2009; Ficklin et al., 2012; Sun et al., 2015) as those estimated for hydrologic models. This suite of performance measures provides a way of assessing agreement between model simulations and observations (NSE), the correlation of the simulated and observed (Pearson R), differences between simulated and observed (RMSE), and percent average difference between simulated and

observed (PBIAS). These statistics provide quantitative measures of model performance and, because of their widespread usage, they also serve as benchmarks for comparison with other models of stream temperature.

In addition, we have modified the method developed by Taylor (2001) for purposes of summarizing model performance based on Pearson R, the correlation coefficient; RMSE, the root-mean-square error between simulated and observed, and RMSD, a measure of the pattern root-mean-square difference. The pattern root-mean-square difference is given as:

$$RMSD = (RMSE^2 + RMSE_{ref}^2 - 2 \cdot RMSE \cdot RMSE_{ref} \cdot R)^{0.5} \qquad (1)$$

where RMSE is the root-mean-square difference of a test field, $RMSE_{ref}$ is a root-mean-square difference of a reference field, and R is the correlation between test and reference fields (Taylor, 2001). In our modification, RMSE is the root-mean-square difference between simulated and observed, $RMSE_{ref}$ is an acceptable measure of standard deviation between simulated and observed, and R is the correlation between simulated and observed. We selected a value of $RMSE_{ref} = 2.0°C$ as a reasonable

measure based on a review of statistical measures found in other water temperature studies (Yearsley, 2012). The goal of our modification is to show how well the model results compare to the reference value for which the RMSE is 2.0°C and the correlation between simulated and observed is equal to 1.0.

### 3.4. Scenarios of Watershed Management

There can be significant impacts on streamflow and stream temperatures in watersheds where there is a transition from forested

to urbanized lands and where there is development of watershed projects for water supply, flood control and generation of hydroelectric power. These are processes that have been ongoing in the Farmington River basin since the 18[th] century and are



likely to continue. As a result there will be a need for watershed planning and models such as the one we have developed that can provide support for these plans. The State of Connecticut's Department of Energy and Environmental Protection, in their Triennial Review of Water Quality Standards (2014), concluded that

"The current temperature criteria in the WQS (Water Quality Standards) are insufficient to be protective of cold water aquatic

life species. Temperature criteria should be revised to better align with fisheries management and restoration priorities of the state".

Beauchene et al. (2014) describe a stream classification system in Connecticut for fishes in three Thermal Classes, "Cold", "Cool" and "Warm" that could be the basis for developing new water temperature criteria for the State of Connecticut's water quality standards. We compare water temperatures simulated with DHSVM-RBM modeling system to the metrics from their

classification system for selected scenarios of management actions. For this example, we consider the following four scenarios in the Farmington River basin that characterize maximum impacts for the riparian vegetation and watershed projects for water supply, flood control and generation of hydroelectric power:

**Scenario 1 (baseline)**: Simulated stream temperatures for the existing conditions for riparian vegetation and water storage projects.

**Scenario 2 (dam removal)**: Simulated stream temperatures for the existing conditions of riparian vegetation and no water storage projects.

**Scenario 3 (riparian removal)**: Simulated stream temperatures for the removal of all riparian vegetation with existing water storage projects remaining in place.

**Scenario 4 (dam & riparian removal)**: Simulated stream temperatures for the removal of both existing riparian vegetation

and water storage projects.

## 4. Results

### 4.1 Hydrology

We simulated streamflows and the energy budget components with DHSVM, for the Farmington River basin by estimating parameters in a manner similar to that described in previous studies (Cao et al., 2016; Cuo et al., 2011; Meyer et al., 1997; Sun

et al., 2015, 2016). Some parameters were estimated from available data sources including physiographic features of the basin. For others, we approached the problem by (1) choosing the range of parameter estimates from previous studies, and (2) varying a subset of parameters until a satisfactory measure of model performance, using the Nash-Sutcliffe Efficiency, NSE, was met.

In our evaluation of DHSVM (Table 4; Figure 3 and 4), we compare model results for both daily and monthly streamflows at six USGS gages distributed across the river network (Figure 1) with the general performance ratings recommended in Table 3





of Moriasi et al. (2007). Although the parameters were calibrated to observed streamflows at USGS gage 01189995 near the basin outlet, the model performance at upstream gages is generally satisfactory. With the exception of daily flows at USGS gage 01187300 and monthly flows at USGS 01185500, both located in smaller tributaries, our results fall within the ratings of Satisfactory (S) to Very Good (VG), recommended in Moriasi et al. (2007). Model performance is best for the two gages with

the greatest drainage area (USGS gage 01189995 and USGSG gage 01188090). We have also included the cumulative distribution function (CDF) for the daily flows as an additional metric for evaluating streamflow results. The CDF's at the two gaging stations with the greatest drainage characterize the distribution of flows better than for those gaging stations in the smaller tributaries. Our simulations do not capture the low flows as well as for the higher flows in the smaller tributaries.

## 4.2 Stream Temperature

We obtained realizations of water temperature using the state estimates for the nominal solution (Supplementary Material) in stream networks for the Farmington River basin (Figure 1). We accessed stream temperature data from the Spatial Hydro-Ecological Decision System (SHEDS) (2016) database at 22 monitoring sites (Figure 1), made by the Connecticut Department of Energy and Environmental Protection (CTDEEP) (2011) for three-hourly times steps in the Farmington River basin, with which to evaluate our simulation results (Figure 5). From this set of observations we calculated the metrics for evaluating

model performance based on the rationale described in Section 3.3, above. The statistical measures that characterize the stream temperature model performance at three-hourly time steps are shown in Table 5 and the modified Taylor diagram for summarizing the degree to which the results cluster near the reference value is shown in Figure 6.

## 4.3. Scenarios of Watershed Management

The four scenarios described in Section 3.4 provide an envelope of stream temperature outcomes for aspects of watershed

management. We simulated stream temperatures in the Farmington River basin within the context of these scenarios. We use the results to examine: (1) the difference in the stream temperature regime in streams between deep reservoirs and those without, and (2) the effect of watershed management on aquatic habitat.

To examine the effect of deep reservoirs on stream temperature, we used the simulations to estimate the average water temperature in the West Branch from its headwaters to the confluence with the Connecticut River for each month for the period

1996-2011. Figure 7 displays these results for the months of April, June, August and October with (Scenario 1) and without (Scenario 2) the two major reservoirs, respectively, and also displays the available monthly-averaged CTDEEP observations for comparison with the actual data.

We made use of the study by Beauchene et al. (2014) to examine the effect of watershed management on aquatic habitat. They defined three stream temperature metrics that classify streams in Connecticut as Cold, Cool, or Warm based on a

comprehensive study of water temperature and fish communities in the rivers of Connecticut, including those in the Farmington River basin. For each of the four management scenarios, we calculated the three metrics, June-August mean water temperature,



July mean water temperature and maximum daily mean water temperature for the period 1996-2011. Based on the results, we estimated the manner in which each of the scenarios would affect thermal habitat for aquatic species throughout the Farmington River basin. Similar to their approach, we estimated the stream temperature classification when two or more of the three conditions in their Table 3 were met. The results are shown in Figures 8(a-d) and Table 6.

Viewed within the context of the thermal classes defined by Beauchene et al. (2014), 6% of the Farmington River basin under present conditions (Scenario1) would be characterized as cold-water habitat, 84% as cool water habitat and 10% as warm water habitat (Table 6). Most of the cold-water habitat is associated with the portion of the Farmington River downstream from the two deep reservoirs, Lake Colebrook and West Branch.

For the unimpounded river (Scenario 2), there is no longer coldwater habitat associated with the deep reservoirs (Figure 8b).
There is only an estimated 2% of coldwater habitat in the basin associated with a few headwaters of the small streams. The percentage of cool water habitat remains the same at 84% (Table 6) and there is a slight increase in the warm water habitat to 14%. Major changes in habitat from cool water to warm water are associated with the simulated results in which riparian vegetation is removed (Scenarios 3 and 4). Cool water habitat is reduced to 44% in Scenario 3 and to 41% in Scenario 4. Warm water habitat increases to 54% in Scenario 3 and 58% in Scenario 4.

## 15 5. Discussion

### 5.1 Hydrology

The version of DHSVM, we used does not have the capability for simulating the effects of water management projects in the watersheds that regulate streamflows and, as a result, alter the natural hydrograph. There are five reservoirs in the basin for which water management for flood control, water supply, fisheries releases and generation of hydroelectric power regulate the
streamflow (Table 1). Two of the largest reservoirs in the basin, Barkhamstead and Nepaug, are sources of drinking water for the City of Hartford, Connecticut for which there is a permitted withdrawal rate of 3.34 m$^3$/sec (120 cfs). Inflow to these two reservoirs can be diverted for this purpose and is not accounted for in the water balance simulated by our version of DHSVM. One of the other deep reservoirs, Lake Colebrook, is an US Army Corps of Engineers project whose primary function is flood control. However, it also provides flow augmentation for fisheries and storage for potential water supply for the City of
Hartford. Inflow is similar to outflow except during periods when the reservoir is drafting for flood storage. For the two other reservoirs treated as deep reservoirs, Lake McDonough and West Branch, outflow is generally equal to inflow and reservoir elevations are relatively constant.

The water management projects we have included in our study (Table 1) modify the natural streamflow to varying degrees. For example, those that are primarily for purposes of municipal water supply such as the Nepaug and the Barkhamstead
Reservoirs divert water that is not accounted for in our analysis. Colebrook Lake is managed for flood control, hydroelectric power generation and riverflow augmentation for fisheries enhancement, all of which can modify the timing and amount of



streamflow. The statistical metrics we calculated (Table 4) are for daily-averaged and monthly-averaged simulated unregulated streamflows for the 15-year period from 1996 through 2011. Nevertheless, the statistical metrics for the simulated unregulated flows at the two stream gages downstream from the major confluences (USGS gages 01188090 and 01189995) are comparable to those obtained in other applications of DHSVM (Cao et al., 2016) and the Soil & Water

Assessment Tool, SWAT (Ficklin et al., 2012). With the exception of $R^2$ at USGS gage 01188090, the results can be characterized as varying from Satisfactory to Very Good using the recommendations of Moriasi et al. (2007).

## 5.2 Stream Temperature

The information required for complete development of the DHSVM-RBM model system in applications of the state-space estimation paradigm is considerable and depends on the time and space scales that are to be modeled, as well as on the

important physical processes that affect stream water temperatures in the basin. For the Farmington River, the spatial resolution of these models is 150 meters in a watershed of 1,559 km$^2$ and the temporal resolution is three hourly for the simulation period, 1996-2011. In addition, there are many processes that affect stream temperature which have received little or no study in the river basin. This includes the effects of groundwater return and hyporheic flow, conduction of heat through the stream bed and snowmelt (e.g., Evans et al., 1998; Kurylyk et al., 2016).

It is not difficult to create the algorithms that incorporate these processes into the state space structure of RBM. However, there is a limited number of stream temperature monitoring sites in SHEDS data base for the Farmington River basin. As a result, the number of degrees of freedom in the parameter space is simply too large to bound the problem with existing observations. Given the size of the parameter space and the limited number of stream temperature monitoring sites, we developed the stream temperature model, RBM with state-space structure within the framework of conservation laws for mass, momentum and

energy. Within this framework, we made use of the results from previous applications (Cao et al., 2016; Sun et al., 2015, 2016) and other studies (e.g., (Boyd and Kasper, 2007; Chapra et al., 2008; Mohseni et al., 1998) to populate the parameter space. We used publicly available databases for land cover (NLCD), surface elevation (NED), soil characteristics (STATSGO), and meteorologic forcing (Livneh et al., 2013) to populate the parameter space of DHSVM to simulate forcing inputs for RBM in the Farmington River basin. Finally, where parameter estimates were not available, we assumed they could be treated as being

similar to other estimates.

In our study, as well as others (Ficklin et al., 2014; van Vliet et al., 2012), the ability to test how well our approach leads to acceptable model results depends on the quality and quantity of observational data from publicly available sources rather than from monitoring networks maintained by state and federal resource agencies. However, there can be more uncertainty in applying these databases compared to studies for which monitoring programs are designed specifically for model development

(e.g., Glose et al., 2017; Vatland et al., 2015).

For this study, we have estimated the metrics for model evaluation, described in Section 3.3, using 22 sites in the SHEDS database where the CTDEEP has archived observations of water temperature (Figure 1). For the statistical measures in Table



5, the modified Taylor diagram (Figure 6) provides a graphical way of summarizing the results in terms of the correlation coefficient (Pearson R), root-mean-square error (RMSE) and the pattern root-mean-square difference (RMSD). These three metrics are referenced to a standard deviation (REF) of 2.0°C, corresponding to the estimated variance for the uncertainty analysis in Yearsley (2012). The clustering of points within the 2.0°C RMSD arc of the Taylor diagram correspond to those

monitoring sites where model performance as measured by Pearson R, the correlation coefficient and RMSE, the root-mean-square error, is within the range of other studies (e.g. Beaufort et al., 2016; Du et al., 2017; Yearsley, 2012). The three monitoring sites in the Farmington River basin that are outside the 2.0°C RMSD arc (CTDEEP-14484, CTDEEP-16091, and CTDEEP-17365) have markedly lower values of standard deviation and correlation coefficient, as well as having negative values of NSE. These three sites are in locations with physical and hydrologic characteristics similar to the other 19

monitoring sites. Without more observations, it is difficult to conclude from the existing data why the simulations at these three sites differ markedly from the observed compared to the other sites.

The ability of water temperature models to accurately simulate diurnal variations in stream temperature increases in difficulty with decreasing the stream depths and speeds. In the DHSVM-RBM model system, we account for this condition by imposing minimum values on stream depths and speed rather than accounting for other depth-dependent process such as exchange of

heat with the stream bed (Boyd and Kasper, 2007; Kurylyk et al., 2016). There are limitations of this approach for conditions of summer low flow periods in freely flowing stream segments of the Farmington River basin. This is evident in the simulated results shown in Figure 5, which overestimate the diurnal variations at some monitoring sites (CTDEEP-14197, CTDEEP-14435, CTDEEP-14484, CTDEEP-14841, CTDEEP-15240, CTDEEP-16061, CTDEEP-17338, CTDEEP-17364 & CTDEEP-17365).

By incorporating two-layer reservoir model (Figure 2) into the RBM model framework, we are able to capture certain important features of the water temperature regime downstream from the two deep reservoirs on the West Branch, Lake Colebrook and West Branch. Figure 9 illustrates the way in which the two-layer model simulates both the delay in timing of the maximum temperature and its magnitude, compared to the simulated results without the reservoirs. Furthermore, there is also a noticeable reduction in the magnitude of the diurnal variations, a feature associated with the thermal inertia of the large volume of water

in the reservoir hypolimnion. The longitudinal distribution of monthly-averaged stream temperatures in the West Branch of the Farmington River (Figure 7) tells a similar story regarding the impact of the deep reservoirs. The impact on stream temperature downstream of reservoirs with long residence times and which have the capability for storing and releasing water from the hypolimnion is evident for the segment of the river below the Lake Colebrook-Hogback series of reservoirs (~River KM 60-95). For the months of April, June and August, the effect of the reservoirs is to provide cooling of stream temperatures

for this segment of the river. The results for October (Figure 7(Oct)) do indicate, however, that the two-layer representation of the reservoirs does not entirely capture the delayed heating effects that are associated with reservoir operations.

Important features of thermal habitat in the Farmington River that emerge from our study include: (1) the importance of headwaters temperatures, (2) the impact of releases from deep reservoirs on downstream water temperatures, and (3) the role





of riparian vegetation. Using the metrics proposed by Beauchene et al. (2014), we have characterized stream segments in the Farmington River basin as primarily cool water habitat (84%) with fewer segments of warm water (10%) and cold water (6%) habitats. In comparison, Beauchene et al. (2014) found that in the 160 sites in Connecticut they studied, 68.1% could be characterized as cool water habitat. Since they do not report the percentages of stream segments with cold water or warm water

habitat, there is no direct comparison with those habitat types. In addition, they limited their study to wadable streams only, and did not include stream segments below dams, as we have.

After removal of the dams (Scenario 2), we estimated that only 2% of the stream segments are coldwater habitat. This means that approximately 4% of the coldwater habitat in our analysis is associated with stream segments in the main stem of Farmington River below the two deep reservoirs (Colebrook Lake and West Branch). Other than dams, geographical and

physical influences on coldwater habitat at basin and subbasin scale ($10^2 - 10^3$ km$^2$) and stream segments of $10^3$ m include elevation, air temperature, precipitation, groundwater/surface-water interaction, hyporheic exchange and land use/landcover (Torgerson et al., 2012). Processes included in the development of the DHSVM-RBM model system are elevation, air temperature, precipitation and land use/land cover, but does not include the contribution from groundwater/surface-water interactions. As Beauchene et al. (2014) observe, however, understanding the role of groundwater may be necessary to explain

temperature variations within the watershed that affect cold water habitat, especially during summer low flow.

## 6. Conclusions

Our goals in this research were: (1) develop nominal solutions for state-space estimates of streamflow and water temperature based on first principles, results of previous studies and data available from publicly available sources, and (2) provide examples of the way in which the models might be applied to watershed management.

Our development of a nominal solution for obtaining state estimates of streamflow and stream temperature has many similarities to other approaches formulated in terms of conservation laws. However, we believe it differs conceptually from others by acknowledging the uncertainty, as well as providing a more general paradigm for environmental models, whether they have been characterized as "data-driven/statistical" or "deterministic/process-based'.

We have parameterized the hydrologic model, DHSVM, and the stream temperature model, RBM, with data from publicly

available sources and taken advantage of the mature status of stream temperature modeling based on the conservation laws of mass, momentum and energy. It is encouraging that when we test our models with available observation, our approach results in metrics that are within the range of similar studies.

By obtaining the nominal solutions for streamflow and stream temperature, we have addressed previous concerns regarding the lack of available information for model development by accessing publicly available data and making use of previous

research. There is room for improvement, of course and it is important to emphasize that maintenance and updating of these databases and monitoring programs is essential.





We have presented an example of the way in which these methods can be used for evaluating watershed management decisions. Our approach to the characterization of aquatic habitat based on simulated thermal classes (Figures 8a-d) should be viewed, first of all, in terms of its potential as a decision support system. Given the robust performance of DHSVM-RBM for the Farmington River basin, the outcomes we have presented provide basic insights into the impacts of land use management on aquatic habitat.

## Acknowledgements

The research on which this paper is based was funded in part by U.S. Environmental Protection Agency Science to Achieve Results (STAR) grant No. R835195 to the University of Washington. The authors gratefully acknowledge the assistance of the Connecticut Department of Energy and Environment in making stream temperature data available. Peter Rochford of Symplectic, LLC in Fairfax, Virginia provided assistance in the implementation of SkillMetrics to produce the modified Taylor diagram. Professor Dennis Lettenmaier of UCLA's Department of Geography provided program guidance during early stages of the study.

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





**Figure 1: Map of Farmington River basin showing (22) CTDEEP water temperature monitoring sites, (6) USGS flow gage sites and (7) major reservoirs characterized by the two-layer module of DHSVM-RBM**





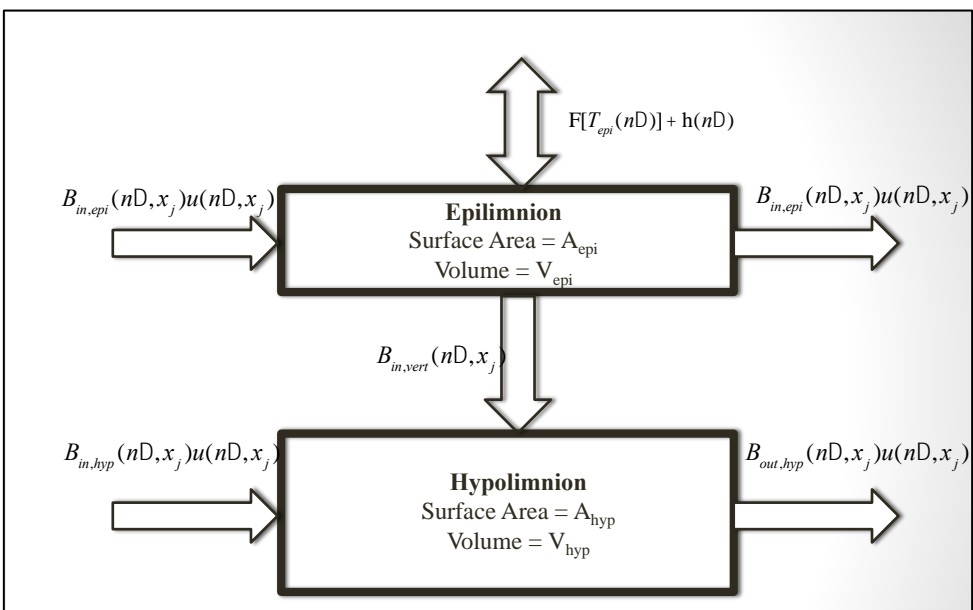

**Figure 2: Conceptual model of Continuously Stirred Reactor (CSTR) for the two-layer reservoir model**





**Figure 3: Time series and cumulative distribution function comparing simulated and observed daily stream flows at selected USGS gages in the Farmington River basin**



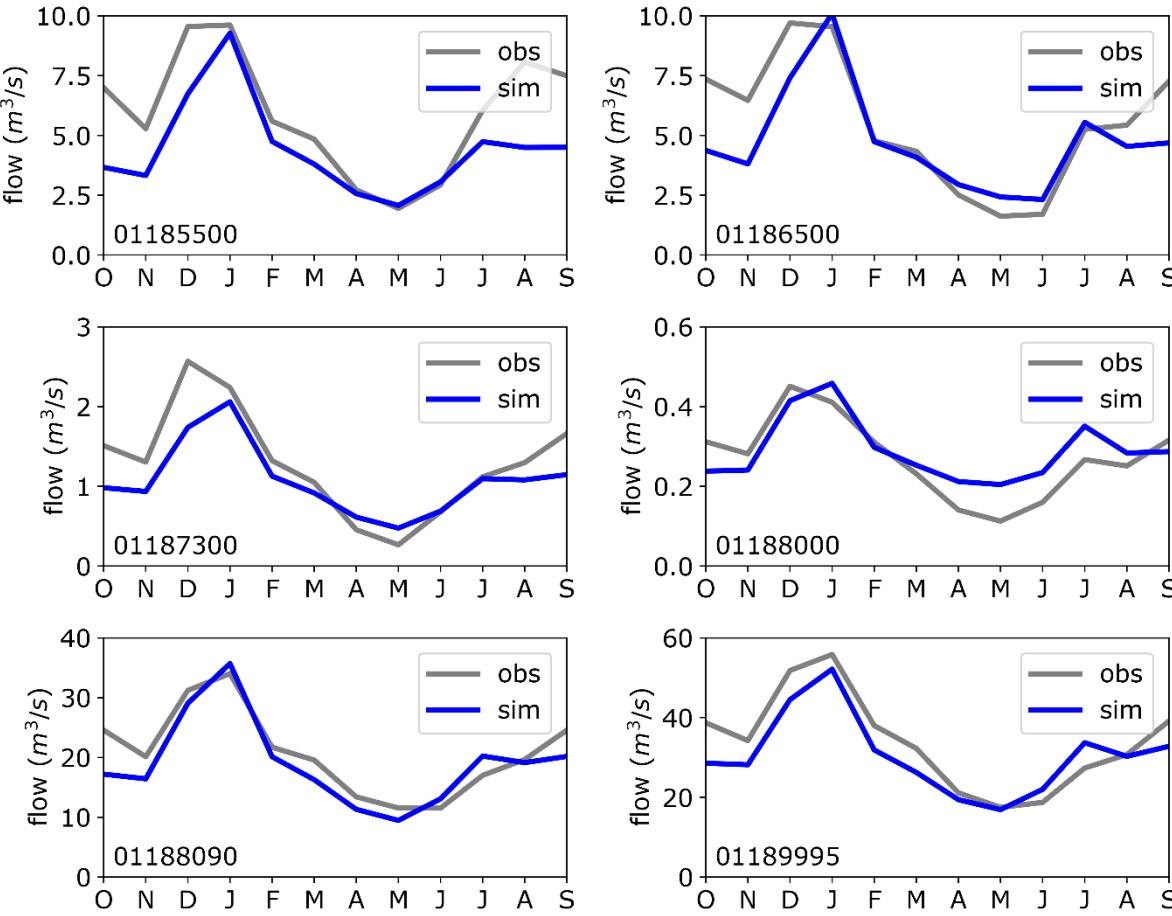

**Figure 4: Time series comparing simulated and observed monthly-averaged stream flows at selected USGS gages in the Farmington River basin**





**Figure 5: Time series comparing simulated and observed three-hourly stream temperatures at selected CTDEEP sites in the Farmington River basin. The period of available temperature observations varies by location. At some locations, the observations were collected for summer months only (e.g. station 15320).**



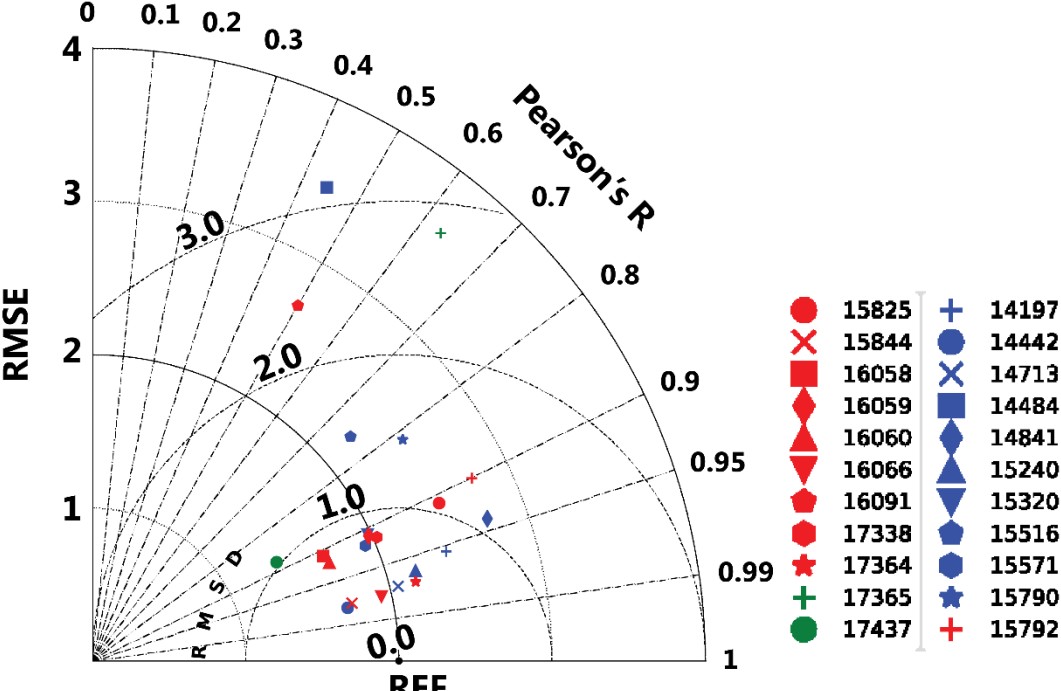

**Figure 6: Modified Taylor diagram (Taylor, 2001) summarizing model performance in terms of RMSE, Pearson R and centered RMSD. The point labeled, "REF", represents a reference value for RMSE from Yearsley (2012).**

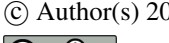



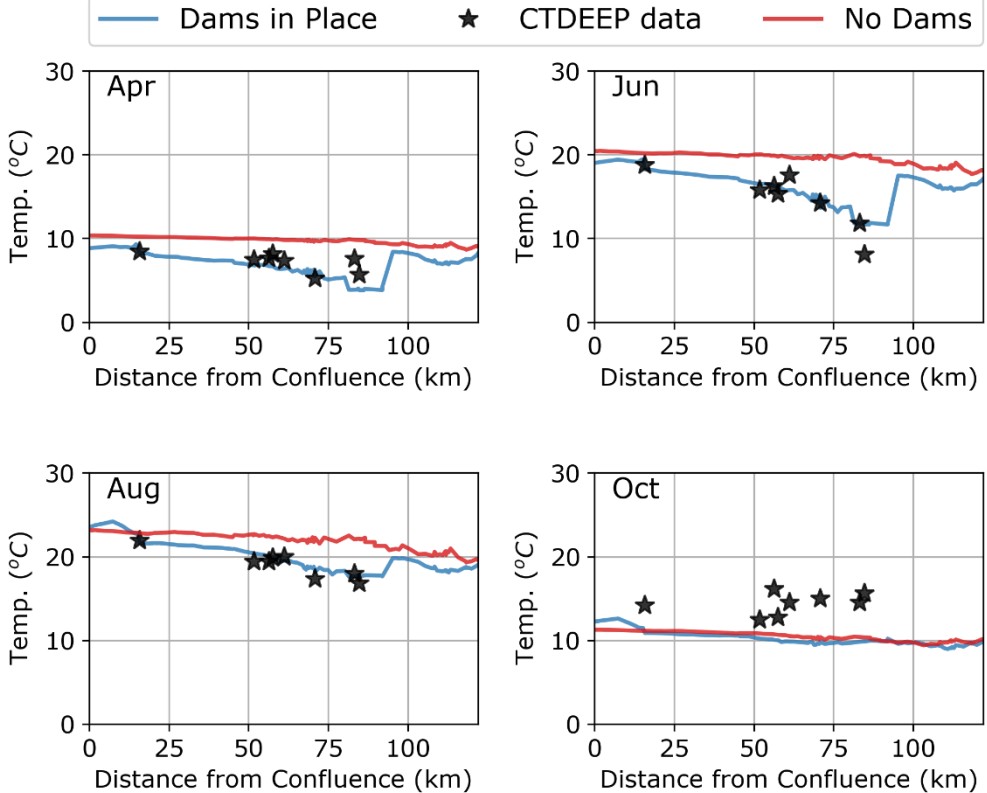

Figure 7: Monthly-averaged stream temperatures of April, June, August and October in the West Branch and Main Stem of the Farmington River for: Dams in Place – Existing riparian shading and water management projects; No Dams – Removal of water management projects maintaining existing riparian shading; and Monthly-averaged water temperatures at locations monitored by the Connecticut Department of Energy and Environment (CTDEEP)





**Figure 8: Thermal classes (cold, cool, and warm) in the Farmington River basin using simulated results for (a) Scenario 1, (b) Scenario 2, (c) Scenario 3, and (d) Scenario 4 from Section 3.4. The green square in (a) and (c) indicate the location of the reservoirs**



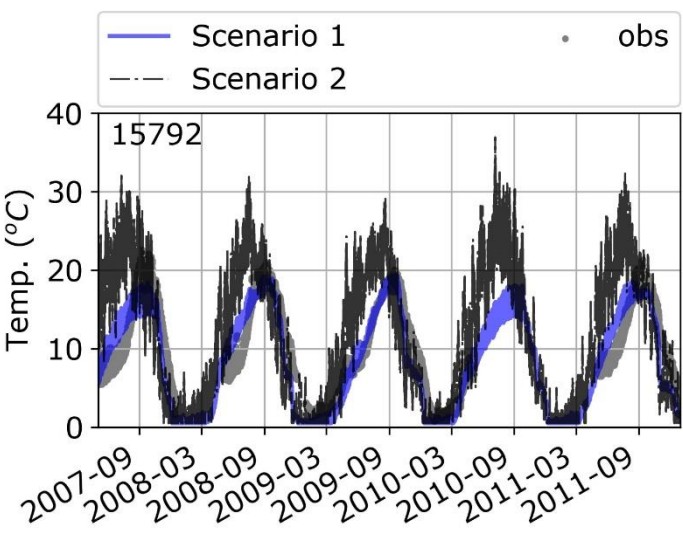

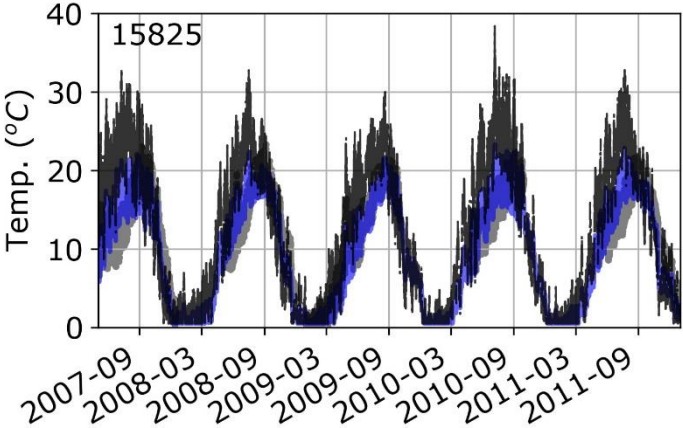

**Figure 9: Simulated and observed water temperatures in the Farmington River downstream from West Branch reservoir (CTDEEP-15792 and CTDEEP 15825)**



**Table 1: Water storage projects in the Farmington River basin modeled with one-dimensional (River Run) or two-dimensional (Deep Reservoir) modules of RBM**

| Reservoir Name (Lat/Lon) | Purpose | Surface Area (m$^2$) | Volume (m$^3$) | Modeled As |
|---|---|---|---|---|
| Otis Reservoir (42.16/-73.06) | Recreation | $4.193 \times 10^6$ | $2.300 \times 10^7$ | River Run |
| Nepaug Reservoir (41.83/-72.94) | Water Supply | $3.592 \times 10^7$ | $1.971 \times 10^8$ | Deep Reservoir |
| Barkhamstead Lake (41.91/- 72.96) | Water Supply | $1.100 \times 10^7$ | $1.390 \times 10^8$ | Deep Reservoir |
| Lake McDonough (41.88/-72.96) | Hydroelectric Power | $1.583 \times 10^6$ | $1.206 \times 10^7$ | Deep Reservoir |
| Colebrook Lake (42.01/-73.04) | Flood Control Hydroelectric Power Fisheries Releases | $2.830 \times 10^6$ | $4.930 \times 10^7$ | Deep Reservoir |
| West Branch (42.00/-73.03) | Hydroelectric Power Fisheries Releases | $8.151 \times 10^5$ | $2.460 \times 10^7$ | Deep Reservoir |
| Rainbow Reservoir (41.92/-72.69) | Hydroelectric Power | $9.490 \times 10^5$ | $5.930 \times 10^6$ | River Run |



**Table 2: Data sets and sources used for estimating parameters in the model identification process**

| Database (Spatial Resolution) | Source | Application |
|---|---|---|
| National Elevation Data Set (NED) (30 meters) | US Geological Survey | DHSVM – Digital Elevation Model (DEM) |
| State Soil Geographic (STATSGO) (1000 meters) | US Department of Agriculture National Resources | DHSVM – Soils Information |
| National Land Cover Database (30 meters) | US Geological Survey | DHSVM – Vegetation Information |
| Meteorological Forcing (1/16° lat. x 1/16° lon.) | Livneh et al. (2013) | DHSVM - Meteorology |
| National Inventory of Dams (NID) Reservoir Area-Volume | US Army Corps of Engineers | RBM – Reservoir Characteristics |
| Reservoir Area-Volume | Julian et al. (2013) | RBM – Reservoir |
| National Water Information System: Web Interface (NWIS) | US Geological Observation | RBM – Leopold Streamflow Data |
| Spatial Hydro-Ecological Decision | SHEDS Development Team | Water Temperature Data |
| Google Maps | Google.com | RBM – Riparian |





**Table 3: Parameters for stream speed and depth in Equations S.7 and S.8 (see Supplementary Materials) estimated from rating measurements at USGS gages in the Farmington River basin. Coefficients from individual USGS gages are extended to other river segments as shown.**

| USGS Gage | Stream Speed | | Stream Depth | | Region of Farmington |
|---|---|---|---|---|---|
| | $U_a$ | $U_b$ | $D_a$ | $D_b$ | |
| West Branch near New 01185500 | 0.0634 | 0.5749 | 0.4175 | 0.2894 | Above Colebrook Lake |
| West Branch Farmington 01186000 | 0.1516 | 0.4342 | 0.3733 | 0.4704 | Hogback Dam to Riverton CT |
| Still River @ 01186500 | 0.0611 | 0.5619 | 0.4631 | 0.3190 | Still River and Tributaries |
| Hubbard River nr West 01187300 | 0.1914 | 0.4253 | 0.5482 | 0.2822 | Watershed above Barkhamstead |
| Bonnell Brook nr 01188000 | 0.3643 | 0.4392 | 0.5482 | 0.2821 | Nepaug Reservoir Watershed |
| Farmington River @ 01188090 | 0.1119 | 0.4237 | 0.1022 | 0.1800 | Riverton CT to Pequaback River |
| Farmington River @ 01189995 | 0.4616 | 0.3203 | 0.1725 | 0.3906 | Pequaback River to Connecticut River |



**Table 4: USGS gage stations in the Farmington River basin (Figure 1) and model performance evaluation criteria for daily and monthly flows simulated by DHSVM based on performance ratings from Moriasi et al. (2007). (VG – Very Good, G – Good, S – Satisfactory, U – Unsatisfactory)**

| USGS Gage | Area (km$^2$) | NSE | | PBIAS | | R$^2$ | |
|---|---|---|---|---|---|---|---|
| | | Daily | Monthly | Daily | Monthly | Daily | Monthly |
| West Branch near 01185500 | 237 | 0.52(S) | 0.33(U) | -25.2(U) | -25.5(U) | 0.57(U) | 0.71(S) |
| Still River @ 01186500 | 220 | 0.58(S) | 0.63(S) | -13.6(G) | -16.9(S) | 0.61(S) | 0.87(VG) |
| Hubbard River nr 01187300 | 52 | 0.48(U) | 0.68(G) | -16.8(S) | -13.6(G) | 0.61(G) | 0.71(S) |
| Bonnell Brook nr 01188000 | 10.6 | 0.54(S) | 0.65(G) | 7.1(VG) | -9.6(VG) | 0.75(S) | 0.85(VG) |
| Farmington River 01188090 | 979 | 0.64(S) | 0.77(G) | -8.4(VG) | 7.1(VG) | 0.54(U) | 0.70(S) |
| Farmington River 01189995 | 1494 | 0.71(G) | 0.76(G) | -9.7(VG) | -8.3(VG) | 0.67(S) | 0.84(G) |



**Table 5: Statistical measures for comparison of daily-averaged simulated and observed water temperatures in the Farmington River**

| | CTDEEP Station | NSE | Pearson's R | BIAS (°C) | RMS (°C) | RMSD |
|---|---|---|---|---|---|---|
| 1 | **14197** Farmington River @ Route 4 | 0.81 | 0.93 | -0.8 | 2.42 | 0.78 |
| 2 | **14435** Salmon Brook adjacent Granbrook Park | 0.46 | 0.95 | 1.67 | 2.50 | 0.86 |
| 3 | **14442** Sandy Brook opposite Grange hall | 0.95 | 0.98 | 0.5 | 1.70 | 0.48 |
| 4 | **14713** Still River@ Route 8 | 0.93 | 0.97 | -0.8 | 2.05 | 0.49 |
| 5 | **14484** West Br.Salmon Brook@ Barndoor Rd | -1.1 | 0.44 | 1.8 | 345 | 3.13 |
| 6 | **14841** West Branch Salmon Brook | 0.78 | 0.92 | -0.7 | 2.74 | 1.10 |
| 7 | **15240** Hubbard Brook u/s Route 20 | 0.86 | 0.96 | 1.2 | 2.19 | 0.60 |
| 8 | **15320** Nod Brook @Footbridge Pond | 0.71 | 0.91 | 0.4 | 1.98 | 0.85 |
| 9 | **15516** Valley Brook d/s Route 20 | 0.06 | 0.77 | 1.6 | 2.23 | 1.50 |
| 10 | **15571** Cherry Brook u/s Route 44 | 0.66 | 0.92 | 1.0 | 1.93 | 0.79 |
| 11 | **15790** Still River @ White St. Bridge | 0.39 | 0.81 | -0.8 | 2.49 | 1.45 |
| 12 | **15792** West Branch Farmington River @ Route 20 | 0.52 | 0.8 | -1.1 | 2.75 | 1.29 |
| 13 | **15825** West Branch Farmington River u/s Route 318 | 0.69 | 0.89 | -1.4 | 2.49 | 1.07 |
| 14 | **15844** Still River u/s Route 20 | 0.95 | 0.98 | -0.3 | 1.74 | 0.49 |
| 15 | **16058** Sandy Brook @ Wolford Hill Rd | 0.78 | 0.91 | 0.6 | 1.65 | 0.85 |
| 16 | **16059** Sandy Brook @ Route 183 | 0.57 | 0.91 | -1.4 | 1.98 | 0.84 |
| 17 | **16060** Sandy Brook @Route 8 | 0.73 | 0.92 | 0.1 | 1.67 | 0.79 |
| 18 | **16066** Farmington River @ Route 189 | 0.85 | 0.97 | -0.6 | 1.93 | 0.44 |
| 19 | **16091** Farmington River nr Bunnell Brook | -1.57 | 0.5 | 0.9 | 2.68 | 2.41 |
| 20 | **17338** Hop Brook @ US Route 10 | 0.69 | 0.92 | 0.8 | 2.03 | 0.82 |
| 21 | **17364** Farmington River nr US Route 4 | 0.82 | 0.96 | -0.6 | 2.17 | 0.53 |
| 22 | **17365** Farmington River nr Burlington | -0.89 | 0.63 | 2.5 | 3.60 | 2.81 |



**Table 6: Percentage of watershed with Cold, Cool or Warm thermal class for watershed management scenarios described in Section 3.4. Metrics for estimating thermal classes are from Beauchene et al. (2014)**

| Habitat Type | Scenario 1 | Scenario 2 | Scenario 3 | Scenario 4 |
|:---:|:---:|:---:|:---:|:---:|
| Cold | 6% | 2% | 2% | 1% |
| Cool | 84% | 84% | 44% | 41% |
| Warm | 10% | 14% | 54% | 58% |