# Peer review of "Assessing the Impacts of Hydrologic and Land Use Alterations on Water Temperature in the Farmington River Basin in Connecticut"

_Hydrology and Earth System Sciences, 2019_

## Referee Comment (RC1) · Anonymous Referee #1 · 3 Jun 2019

As outlined by the authors, the study aims to better predict the impact of reservoirs and riparian cover on water temperature using the coupled DHSVM-RMB modeling system and to assess the usefulness of the modeling system to aid in watershed planning. The authors conducted four simulations (baseline, removal of dams, removal of riparian buffers, and removal of dams and riparian buffers) and evaluated how water temperature dynamics changed across these scenarios. Overall, results indicated that larger reservoirs were providing a cooling effect downstream of outlets and that riparian shading also provided a cooling effect on water temperature. However, there appears to be another aspect of the manuscript which focuses more on the use of state-space models. But this section on state-space models is not currently well aligned with the

reservoir and riparian buffer component of the model and detracts from the readability.

As currently written, I recommend this work to be considered for publication with major modifications as outline below. While the inclusion of the reservoirs in the RMB model is novel, the authors need to make the manuscript considerably more focused to be considered for publication.

General: Per the title, this manuscript appears to use a water temperature model to simulate the impacts of reservoirs and riparian cover on water temperature dynamics. However, the introduction does not appear to cover these topics in much detail, but rather spends most of the text outlining the use of state-space models. While the description of state-space modeling is interesting, this reviewer would like to see the introduction adapted and to focus more on the ways in which reservoirs and riparian cover alter steam temperature dynamics and how this manuscript will address this.

General: The method used to estimate riparian vegetation characteristics (outline on page 4) seem lacking in text detail and potentially very error prone (i.e. someone manually using Google streets to record tree heights for hundreds of miles of stream). The authors state that Google street view was used to estimate canopy height by comparing vegetation to nearby features, such as telephone poles. The reviewer feels this is a very subjective method, which would need some type of validation approach before it should be uses in a published study. Additionally, how to tree height estimates for zones where there is not Google street view available, i.e. headwaters?

General: Pertaining to results shown in Figure 8, where water temperatures are classified into cold, cool, and warm. Why does Scenario 2 (subplot B) having different conditions above reservoirs compared to the baseline condition (subplot A). It seems that some small streams above reservoirs change from being cold in the baseline condition to be cool. It is hard for the reviewer to find a justification for this output. For example, why would removing a reservoir tens of kilometers downstream impact upstream headwater temperatures. This potential error puts into question the other results. Please

consider re-checking the model structure and output to ensure this is not an error.

General: The role of riparian shading seems to be completely missing from the discussion. Since this was one of the two perturbed characteristics of the system, the findings should be discussed.

Specific:

Page 2, paragraph 1: This paragraph seems out of place for the second paragraph in the introduction. The reviewer would prefer to see more background into the topic being addressed in the manuscript before jumping into some of the technical details of the modeling.

Page 3, line 22: Please consider providing some explanation/criteria for why the seven large reservoirs were thought to significantly modify the thermal regime of the basin. For example, do the outlet works of these reservoirs allow water to be drawn from different depths and thus one could have cold hypolimnion water being released during the warm summer period?

Page 4, lines 16 and 17: Please consider indicating what method was used to re-scale the 30 m data to 150 m.

Page 5, line 17: Pleas clarify why only gage CT-15844 was use to fit the relationship and applied to all other gages. How valid is the assumption that these parameters are representative of the headwaters? It would be preferred to see how much variability there is in the parameters across gages to better understand how this variability might impact headwater temperature inputs.

Page 6, line 3: How valid is the assumption that volume remains constant during the simulation period (multiple years)? For example, in reservoirs functioning as a flood control mechanism, one would think that their volume would change over a period of a year. Please consider adding some text to justify this assumption or address how it is a limitation in the modeling in the discussion section.

[Figure]

Page 9, line 23: This paragraph reads like it belongs in the methods sections. Please consider re-ordering.

Page 10, line 3: Text starts with 'similar to their approach'. Please be more specific.

Page 10, line 4: Since the reviewer/reader does not have access to Table 3 in Beauchene et al. (2014), the authors should consider a different way of referencing this table as currently it is not helpful to the reader.

Page 10, line 17: Please consider opening the discussion with a paragraph that better orients the reader to the main goals/methods of the manuscript and the primary findings. As it currently reads, the first paragraph of the discussion seems to point to limitations in the modeling, which would be better suited later in the text.

Page 11, line 8: This paragraph does not belong in the discussion, it is merely restating general ideas about state-space model and water temperature dynamics. Consider placing somewhere other than the discussion or re-write to relate the work performed in the manuscript to other research and future efforts.

Page 11, line 15: Similar comment to Page 11, line 8.

Page 11, line 26: A paragraph needs at least 3 sentences. Additionally, this paragraph seems only partially thought out.

Page 12, line 16: The statement that about diurnal variation is difficult to assess in Figure 5 which sometime spans multiple years (i.e. diurnal variation cannot be seen). Consider generating new plots with shorter time periods if the authors wish to discuss this modeling issue in the discussion.

---

## Referee Comment (RC2) · Anonymous Referee #2 · 23 Aug 2019

General comments

The manuscript documents the extension of the coupled hydrologic and stream temperature models DHSVM-RBM to include the simulation of the effect of stratified and "run-of-river" reservoirs on stream temperature. The riparian vegetation model was also modified to account for changes in riparian vegetation in time and space, although it seems this modification was not evaluated in this study. Simplifying assumptions were acknowledged (e.g., constant water elevation in reservoirs, no water diversions, no river exchange with groundwater or bed heat exchange) and potential for further improvement of the model was recognized. In general, the authors describe the further

development of the DHSVM-RBM modeling system and demonstrate its capabilities using readily available data for the Farmington River basin. In particular, the model evaluation (error statistics and graphical presentations) is well done and highlights what the model did relatively well (e.g., high river flows) and what it did not (e.g., low flows, stream temperature at headwater locations). Some statement might be needed to explain the potential effect of low flow prediction errors on temperature.

The manuscript appears to assume that the reader is already familiar with the complexities of stream temperature modeling and the potential effects of reservoirs on downstream temperature. For example, there is no mention of the potential difference in effects of surface vs deep reservoir outlets, although the main effect described is downstream cooling due to "storing and releasing water from the hypolimnion" (line Discussion p. 12, line 27). The need for time and space varying riparian vegetation in the model was also not explained and there is not much discussion of the effects of riparian loss on water temperatures.

I believe the manuscript would benefit from at least a brief section on stream temperature modeling with an emphasis on riparian shade effects and the potential effects of various types of reservoirs on downstream temperatures. Some discussion of the effect of riparian loss on temperature results

Specific comments

p. 6, line 5: Recommend including the location of the outlet for each reservoir in Table 1 or Table 1

p. 6, line 8: Niemeyer et al. (2018) does not appear in references

p. 9, line 4: Moriasi et al. (2007) does not appear in references

p. 10, line 16, 5.1 Hydrology: From these two paragraphs it appears that this version of DHSVM does not account for transient storage or water diversions, which was implied earlier by stating that all reservoir inflows were equal to outflows. I wasn't until p.

11, line 2 that I understood that the flow model evaluation was based on long-term naturalized flows. It might help the reader to make this point in 3.3 Model Evaluation section. It was also confusing that the flow calibration figures and text refer to observed rather than naturalized flows. I'm assuming they are the latter.

p. 11, line 26: I'm not sure I understand the purpose of this paragraph. I thought the temperature data were collected by a Connecticut government agency. Furthermore, the application was a demonstration so would naturally rely on available data. Furthermore, in practical applications, I think it is rare for modelers to design the data collection systems to support model development – for flow networks in particular.

p. 12, line 30: The inability of the model to simulate the delayed warming effect of deep reservoir releases seems like a significant weakness in the model. It might be useful to suggest how the model might be improved to better simulate this feature.

p. 13, line 9: The general influences on water temperature might be a good starting point for material in the introduction describing basic stream temperature modeling and the role riparian vegetation and reservoirs modify temperature.

---

## Author Comment (AC1) · 18 Sep 2019

**Responses to comments posted by Referee #1**

As outlined by the authors, the study aims to better predict the impact of reservoirs and riparian cover on water temperature using the coupled DHSVM-RMB modeling system and to assess the usefulness of the modeling system to aid in watershed planning. The authors conducted four simulations (baseline, removal of dams, removal of riparian buffers, and removal of dams and riparian buffers) and evaluated how water temperature dynamics changed across these scenarios. Overall, results indicated that larger reservoirs were providing a cooling effect downstream of outlets and that riparian shading also provided a cooling effect on water temperature. However, there appears to be another aspect of the manuscript which focuses more on the use of state-space models. But this section on state-space models is not currently well aligned with the reservoir and riparian buffer component of the model and detracts from the readability. As currently written, I recommend this work to be considered for publication with major modifications as outline below. While the inclusion of the reservoirs in the RMB model is novel, the authors need to make the manuscript considerably more focused to be considered for publication.

**Response**: We thank the reviewer for constructive comments that have been of great help in improving the manuscript. Our responses to the comments are given below.

**General**: Per the title, this manuscript appears to use a water temperature model to simulate the impacts of reservoirs and riparian cover on water temperature dynamics. However, the introduction does not appear to cover these topics in much detail, but rather spends most of the text outlining the use of state-space models. While the description of state-space modeling is interesting, this reviewer would like to see the introduction adapted and to focus more on the ways in which reservoirs and riparian cover alter steam temperature dynamics and how this manuscript will address this.

**Response**: We have rewritten the Introduction to provide background on the effects of dams and riparian vegetation on stream temperature. We agree with the reviewer that the paper will benefit from more discussion of the impacts of hydrologic modification and loss of riparian shading and have added relevant text in Sections 4.3 (pg 12, lines 3-23) and Section 5.2 (pg 15-16). However, we would also like to ensure that our concept of model development is considered. We believe that state-space estimation represents an approach to model development that is more general than that of characterizing models as either process/deterministic or statistical and have therefore maintained this discussion and revised the text throughout the manuscript to tie this better to the rest of the paper.

**General**: The method used to estimate riparian vegetation characteristics (outline on page 4) seem lacking in text detail and potentially very error prone (i.e. someone manually using Google streets to record tree heights for hundreds of miles of stream). The authors state that Google street view was used to estimate canopy height by comparing vegetation to nearby features, such as telephone poles. The reviewer feels this is a very subjective method, which would need some type of validation approach before it should be uses in a published study. Additionally, how to tree height estimates for zones where there is not Google street view available, i.e. headwaters?

**Response**: As seen in a large body of literature, it is a common practice to apply basin or subbasin uniform riparian parameters based on 'representative' tree species in basin-scale temperature modeling where field data on riparian vegetation are often lacking. In comparison, the survey approach we adopted here combined with literature values and the results from previous studies is, in fact, much less error-prone. We have incorporated our responses to the

revised text in Section 3.1 (pg 5, lines 18-32).

We inferred vegetation types (trees and shrubs) from forest surveys (Wharton et al., 2004), and assigned tree height based on tree species (Wharton et al., 2004; Lamson, 1987). In addition, where possible, we compared our derived tree height with utility poles using Google Maps Street View, assuming that the average height of wooden utility poles is 15 meters (https://en.wikipedia.org/wiki/Utility_pole). The survey of riparian vegetation characteristics was performed by Marise Baptise, one of the paper's co-authors. There are approximately 750 km of stream segments in our model of the Farmington River watershed. Ms. Baptiste performed this analysis in less than a week, enabled by the fact that forest coverage in the watershed is reasonably uniform. In Section 3.1 (pg 5, lines 18-32), we have referenced those sources that characterize the forests of Connecticut. We also believe that the use of cultural features such as utility poles provides a reasonable good estimate of tree height and provided a reference.

References:

Lamson, N.I.:  Estimating Northern Red Oak Site-Index Class from Total Height and Diameter of Dominant and Codominant Trees in Central Appalachian Hardwood Stands, NE-RP-605, Northeastern Forest Experiment Station, Forest Service, United States Department of Agriculure, 1987.

Wharton, E. H., Widmann, R. H., Alerich, C. L., Barnet, C. H., Lister, A. J., Lister, T. W., Smith, D. and Borman, F.: The forests of Connecticut. Resour. Bull. NE-160, 2004.

**General**: Pertaining to results shown in Figure 8, where water temperatures are classified into cold, cool, and warm. Why does Scenario 2 (subplot B) having different conditions above reservoirs compared to the baseline condition (subplot A). It seems that some small streams above reservoirs change from being cold in the baseline condition to be cool. It is hard for the reviewer to find a justification for this output. For example, why would removing a reservoir tens of kilometers downstream impact upstream headwater temperatures. This potential error puts into question the other results. Please consider re-checking the model structure and output to ensure this is not an error.

**Response**: Thank you for pointing this out. We did, indeed, have an error in the input for Scenario 2. We corrected the error and updated Figure 8 and Table 7.

**General**: The role of riparian shading seems to be completely missing from the discussion. Since this was one of the two perturbed characteristics of the system, the findings should be discussed.

**Response**: We agree that the paper would benefit from a more thorough discussion of the impact of both reservoirs and riparian shading on stream temperature. In response, we have expanded result analyses in Sections 4.3 (pg 12, lines 3-23) and expanded on discussions in Section 5.2.

**Specific**:
Page 2, paragraph 1: This paragraph seems out of place for the second paragraph in the introduction. The reviewer would prefer to see more background into the topic being addressed in the manuscript before jumping into some of the technical details of the modeling.

**Response**: We have rewritten the Introduction in response to this comment.

Page 3, line 22: Please consider providing some explanation/criteria for why the seven large

reservoirs were thought to significantly modify the thermal regime of the basin. For example, do the outlet works of these reservoirs allow water to be drawn from different depths and thus one could have cold hypolimnion water being released during the warm summer period?

**Response**: We have added discussion to this comment in Section 2 (pg 4, lines 22-30).

Page 4, lines 16 and 17: Please consider indicating what method was used to re-scale the 30 m data to 150 m.

**Responses**: We have added the method for re-scaling to Section 3.1 (pg 6, line 11).

Page 5, line 17: Please clarify why only gage CT-15844 was use to fit the relationship and applied to all other gages. How valid is the assumption that these parameters are representative of the headwaters? It would be preferred to see how much variability there is in the parameters across gages to better understand how this variability might impact headwater temperature inputs.

**Response**: Headwaters play an important role in maintaining coldwater habitat, as we show in Figure 8. We agree it would be valuable to understand how parameter variability affects our results. However, there are four parameters in the Mohseni relationship (Equation S.12) and 160 headwaters. We believe that evaluating the variability would be beyond the scope of this study.

For our study we have access to limited data with multi-year continuous temperature records from only three stations located near headwaters. They are CT-14442, CT-14713, and CT-15844. Based on the model evaluation on temperature predictions (Figure 5), the use of Mohseni parameters derived from CT-15844 for all headwater segments yielded quite good agreement between simulations and observations where available. For example, at CT-14442 and CT-14713 gages, NSE > 0.9 and Pearson's R is close to 1.0. This, to the extent possible, provides support for assuming that the applied parameters are representative and robust for modeling the headwater temperatures in the basin. We have revised the text in Section 3.2 (pg 6, lines 9-15) in response to this comment.

Page 6, line 3: How valid is the assumption that volume remains constant during the simulation period (multiple years)? For example, in reservoirs functioning as a flood control mechanism, one would think that their volume would change over a period of a year. Please consider adding some text to justify this assumption or address how it is a limitation in the modeling in the discussion section.

**Response**: We have added discussion in Section 5.2 in response to this comment (see pg 16, lines 18-25).

Page 9, line 23: This paragraph reads like it belongs in the methods sections. Please consider re-ordering.

**Response**: We moved this paragraph to Section 3 as recommended.

Page 10, line 3: Text starts with 'similar to their approach'. Please be more specific.

**Response**: We modified the text to be more specific, as recommended (pg 12, line 31).

Page 10, line 4: Since the reviewer/reader does not have access to Table 3 in Beauchene et al. (2014), the authors should consider a different way of referencing this table as currently it is not

helpful to the reader.

**Response**: We included Table 3 of Beauchene et al (2014) as Table 6 and renumbered the original Table 6 as Table 7.

Page 10, line 17: Please consider opening the discussion with a paragraph that better orients the reader to the main goals/methods of the manuscript and the primary findings. As it currently reads, the first paragraph of the discussion seems to point to limitations in the modeling, which would be better suited later in the text.

**Response**: We've revised the leading paragraph in Section 5.1 as the reviewer suggested.

Page 11, line 8: This paragraph does not belong in the discussion, it is merely restating general ideas about state-space model and water temperature dynamics. Consider placing somewhere other than the discussion or re-write to relate the work performed in the manuscript to other research and future efforts.

**Response**: We have revised and added text to Section 5.2 (pg 14-16) by including a more general discussion of model related research, model limitations, and future improvements.

Page 11, line 15: Similar comment to Page 11, line 8.

**Response**: We have added to Section 5.2 (pg 14-16) a more general discussion of model related research, model limitations, and future improvements.

Page 11, line 26: A paragraph needs at least 3 sentences. Additionally, this paragraph seems only partially thought out.

**Response**: We have added text to this paragraph and discussed the point that sampling design is not always done with model development in mind. Furthermore, this can lead to more uncertainty in state estimates (pg 15, lines 2-5).

Page 12, line 16: The statement that about diurnal variation is difficult to assess in Figure 5 which sometime spans multiple years (i.e. diurnal variation cannot be seen). Consider generating new plots with shorter time periods if the authors wish to discuss this modeling issue in the discussion.

**Response**: To enhance readability of Figure 5, we kept six CTDEEP sites in Figure 5 and moved the remaining 16 sites to Figure S.1 in Supplementary Material.

[revised manuscript text omitted]

---

## Author Comment (AC2) · 18 Sep 2019

**Responses to comments posted by Referee #2**

**General comments**

The manuscript documents the extension of the coupled hydrologic and stream temperature models DHSVM-RBM to include the simulation of the effect of stratified and "run-of-river" reservoirs on stream temperature. The riparian vegetation model was also modified to account for changes in riparian vegetation in time and space, although it seems this modification was not evaluated in this study. Simplifying assumptions were acknowledged (e.g., constant water elevation in reservoirs, no water diversions, no river exchange with groundwater or bed heat exchange) and potential for further improvement of the model was recognized. In general, the authors describe the further development of the DHSVM-RBM modeling system and demonstrate its capabilities using readily available data for the Farmington River basin. In particular, the model evaluation (error statistics and graphical presentations) is well done and highlights what the model did relatively well (e.g., high river flows) and what it did not (e.g., low flows, stream temperature at headwater locations).

**Response**: We thank the reviewer for the positive feedback. We have addressed the comments in the response below and revised the manuscript accordingly.

Some statement might be needed to explain the potential effect of low flow prediction errors on temperature.

**Response**: We have added text in Section 5.2 (pg 15, lines 20-29) that discusses the effect of low flow prediction errors on temperature modeling.

The manuscript appears to assume that the reader is already familiar with the complexities of stream temperature modeling and the potential effects of reservoirs on downstream temperature. For example, there is no mention of the potential difference in effects of surface vs deep reservoir outlets, although the main effect described is downstream cooling due to "storing and releasing water from the hypolimnion" (line Discussion p. 12, line 27). The need for time and space varying riparian vegetation in the model was also not explained and there is not much discussion of the effects of riparian loss on water temperatures I believe the manuscript would benefit from at least a brief section on stream temperature modeling with an emphasis on riparian shade effects and the potential effects of various types of reservoirs on downstream temperatures. Some discussion of the effect of riparian loss on temperature results.

**Response**: We agree that the paper would benefit from expanded analyses and discussion on the role of both reservoirs and riparian shading. We have rewritten the Introduction that describes many of the aspects of the effect of dams and riparian vegetation on stream temperature. In Section 3.1 (pg 5), we have added text to address the need for time and space varying riparian vegetation in temperature modeling, and elaborated the approach we took to develop the riparian vegetation parameters. In Sections 4.3 (pg 12, lines 3-23) and Sections 5.2 (pg 15-16), we have expanded analyses and discussions on the effect of riparian loss and reservoirs on stream temperature modeling.

**Specific comments**

p. 6, line 5: Recommend including the location of the outlet for each reservoir in Table 1 or Table 1

**Response**: We have added the locations of outlets to Table 1.

p. 6, line 8: Niemeyer et al. (2018) does not appear in references

**Response**: We have added this reference.

p. 9, line 4: Moriasi et al. (2007) does not appear in references

**Response**: We have added this reference.

p. 10, line 16, 5.1 Hydrology: From these two paragraphs it appears that this version of DHSVM does not account for transient storage or water diversions, which was implied earlier by stating that all reservoir inflows were equal to outflows. I wasn't until p.11, line 2 that I understood that the flow model evaluation was based on long-term naturalized flows. It might help the reader to make this point in 3.3 Model Evaluation section. It was also confusing that the flow calibration figures and text refer to observed rather than naturalized flows. I'm assuming they are the latter.

**Response**: The version of DHSVM we used does not account for water management projects. The flows simulated by DHSVM are the unregulated flows, but we calibrated/validated these flows with observed streamflows collected at the USGS flow gages. We have clarified this in Section 3.1 (pg 5, lines 10-16). In addition, we now discuss the effect of our simplified modeling approach (i.e. natural flow without regulation) on temperature modeling, and recommend future model improvements in Section 5.2 (pg 16, lines 18-25).

p. 11, line 26: I'm not sure I understand the purpose of this paragraph. I thought the temperature data were collected by a Connecticut government agency. Furthermore, the application was a demonstration so would naturally rely on available data. Furthermore, in practical applications, I think it is rare for modelers to design the data collection systems to support model development – for flow networks in particular.

**Response**: We have revised text and discussed the point that sampling design is not always done with model development in mind. Furthermore, this can lead to more uncertainty in state estimates (pg 15, lines 1-7).

p. 12, line 30: The inability of the model to simulate the delayed warming effect of deep reservoir releases seems like a significant weakness in the model. It might be useful to suggest how the model might be improved to better simulate this feature.

**Response**: We have added text in Section 5.2 in response to this comment (pg 16, lines 18-25).

p. 13, line 9: The general influences on water temperature might be a good starting point for material in the introduction describing basic stream temperature modeling and the role riparian vegetation and reservoirs modify temperature.

**Response**: We have added text in the Introduction and Section 4.3 and 5.2 which we believe responds to this comment.

[revised manuscript text omitted]